# The Internal Growth Function: A More General PAC Framework for Scenario Decision Making

**Guillaume O. Berger**                                   *guillaume.berger@uclouvain.be*
*Department of Applied Mathematics ICTEAM Institute*
*UCLouvain*

**Raphaël M. Jungers**                                    *raphael.jungers@uclouvain.be*
*Department of Applied Mathematics, ICTEAM Institute*
*UCLouvain*

**Reviewed on OpenReview:** *https://openreview.net/forum?id=HqPKJSAkrp*

## Abstract

This paper introduces a new PAC framework for scenario decision-making problems. Scenario decision making consists in making a decision that satisfies a probabilistic constraint (also called a chance constraint) from finitely many sampled realizations (called scenarios) of the constraint. PAC bounds are sufficient conditions on the number of samples to guarantee with high confidence that the sample-based decision satisfies the true constraint with a prescribed probability. Existing PAC bounds rely on intrinsic properties of the problem, such as convexity (Calafiore and Campi, 2005), finite VC dimension (Alamo et al., 2009) or existence of a compression scheme (Margellos et al., 2014). While powerful in some applications, these PAC bounds can be vacuous (or infinite) when the properties are not satisfied. In this paper, we propose a new PAC framework, leading to PAC bounds that are not vacuous for a strictly larger class of scenario decision-making problems. This bound is based on the novel notion of "internal growth", which adapts the notion of "growth function" from classical machine learning (Vapnik and Chervonenkis, 1968) to scenario decision making. We also relate this notion to other novel properties of the system, such as the $k$-VC dimension. Furthermore, we show a partial converse result: namely, that for the family of stable monotone scenario decision algorithms, the algorithm is PAC if *and only if* it satisfies our criterion. Finally, we demonstrate the usefulness of our framework, and compare with existing approaches, on practical problems.

## 1 Introduction

Many problems in engineering involve making a decision in the face of *random* constraints: e.g., controlling a robot in an environment with randomly moving obstacles (Campi & Garatti, 2008), planning energy production under uncertain demand (Antoniadou-Plytaria et al., 2022), portfolio management with prescribed risk tolerance (Pagnoncelli et al., 2012), smart building management with safety and comfort requirements (Parisio et al., 2014), machine learning (Campi & Garatti, 2023), etc. The probability that a decision violates a given random constraint is called its *risk*. Making a decision that has a low risk is challenging for two reasons: (i) the constraint landscape is often intractable, as it typically consists in infinitely many possible realizations of the constraint with the non-convex requirement that the decision satisfies a "high-probability" subset of these realizations (Blackmore et al., 2010; Mammarella et al., 2022); (ii) the distribution of the chance constraint is often unknown in practical applications, which makes a model-based design impossible.

The approach of *scenario decision making* (Campi & Garatti, 2008; Alamo et al., 2009; Campi & Garatti, 2011; Margellos et al., 2014; Esfahani et al., 2015; Campi & Garatti, 2018; Campi et al., 2018; Garatti & Campi, 2022; Campi & Garatti, 2023; Rocchetta et al., 2024) provides a simple and effective way to address

these challenges by leveraging the principle of sample-based methods. The approach consists in drawing $m$ sampled realizations (called *scenarios*) of the random constraint, and making a decision that satisfies all, or a predefined fraction, of the sampled constraints. The decision made from the scenarios is called the *scenario decision*, and the function mapping the $m$ scenarios to their associated scenario decision is called the *scenario decision algorithm*. A key aspect of this approach is that under some assumptions on the problem or algorithm, one can provide guarantees on the risk of the scenario decision, called *PAC* (*Probably Approximately Correct*) guarantees (Shalev-Shwartz & Ben-David, 2014; Mohri et al., 2018): that is, upper bounds on the probability (with respect to the sampling of the $m$ scenarios) that the risk of the scenario decision exceeds some prescribed tolerance $\epsilon$. These bounds are referred to as *confidence bounds.* Existing PAC guarantees for scenario decision algorithms are complexity-based (De Farias & Van Roy, 2004; Alamo et al., 2009; Lauer, 2024), or compression-based (Campi & Garatti, 2008; Calafiore, 2010; Campi & Garatti, 2023; Margellos et al., 2015), following the classification in the recent survey by Rocchetta et al. (2024). Complexity-based guarantees rely on the identification of decisions as binary classifiers of the constraints (between those that are satisfied by the decision, and those that are violated), and leverage classical results in the theory of binary classification learning, such as the VC dimension (Vapnik & Chervonenkis, 1971), which provide necessary and sufficient conditions for a set of classifiers to be PAC-learnable (Shalev-Shwartz & Ben-David, 2014; Mohri et al., 2018). Compression-based guarantees, which apply among others to convex scenario optimization problems (Campi & Garatti, 2008; Margellos et al., 2014), rely on the property that any set of scenarios can be compressed to a subset of bounded size (called the compression size) and give the same decision as the original set of scenarios. However, some scenario decision algorithms do not satisfy any of these properties (Berger & Jungers, 2025), motivating the need for new PAC guarantees for scenario decision algorithms.

In this paper, we propose a new type of PAC guarantees that is strictly more general than the PAC guarantees mentioned above. These PAC guarantees rely on a novel notion called the *internal growth function* of the scenario decision algorithm, and is inspired from the notion of *growth function* in binary classification learning (Shalev-Shwartz & Ben-David, 2014; Mohri et al., 2018) (Section 3). We show that an internal growth function that is sub-exponential (called *slow internal growth*) translates to PAC guarantees for the scenario decision algorithm (Section 4). Furthermore, we relate the slow internal growth property to the criteria mentioned above, showing that it generalizes them, and to another novel quantity, called the *k-VC dimension*, which refines the notion of VC dimension for scenario decision algorithms (Section 5). Additionally, we present partial converse results: we show that for the class of *stable monotone* scenario decision algorithms (studied, e.g., in the context of one-class classification learning; Eppstein & Erickson, 1994; Chan & Har-Peled, 2021), the slow internal growth property is sufficient *and necessary* to obtain PAC guarantees (Section 6). Finally, we demonstrate the usefulness of the new PAC guarantees on concrete examples of scenario optimization problems, where the existing approaches fail to show the PAC property, but our approach succeeds (Section 7).

**Related works**

The recent survey by Rocchetta et al. (2024) classifies PAC bounds into three categories: (i) *complexity-based PAC bounds* (De Farias & Van Roy, 2004; Alamo et al., 2009; Lauer, 2024), based namely on the notions of VC dimension or Rademacher complexity; (ii) *compression-based PAC bounds* (Littlestone & Warmuth, 1986; Floyd & Warmuth, 1995; Margellos et al., 2015), based on the notion of compression; and (iii) *scenario-based PAC bounds* (Campi & Garatti, 2008; Calafiore, 2010; Campi & Garatti, 2023), that are tailored to convex scenario optimization problems (and sometimes exploit additional assumptions, like "non-degeneracy" or "non-concentrated mass" to obtain better bounds; Campi & Garatti, 2023). A close connection between compression-based and scenario-based PAC bounds was established in Margellos et al. (2015), showing among others that scenario-based PAC bounds imply the existence of a compression scheme for the scenario decision algorithm, and thereby can be also analyzed in this way.

In a recent paper (Berger & Jungers, 2025), we show that complexity-based and compression-based PAC bounds are incomparable in scenario decision making (contrarily to the classical classification problem; Moran & Yehudayoff, 2016), in the sense that one can be vacuous for some scenario decision algorithms, while the other remains informative, and vice-versa. As a corollary, it follows that the underlying properties (i.e.,

finite VC dimension or existence of a compression scheme) are *not* necessary to obtain PAC guarantees for scenario decision algorithms. The criterion based on the internal growth proposed in this paper is more general in that it is satisfied whenever any of the aforementioned properties is satisfied, and is also shown to be satisfied for some scenario decision algorithms, even when none the other properties are. Furthermore, for monotone algorithms, it is shown to be satisfied if and only if the algorithm has PAC guarantees. The question of whether this reverse direction holds for all scenario decision algorithms remains open.

The abstract view on scenario decision algorithms considered in this work (Section 2.2) draws a connection with the problem of *one-class classification* where one tries to learn the support of a distribution (see, e.g., Perera et al., 2021 for a survey). However, research in this area mainly focuses on analyzing specific algorithms (e.g., SVM-based or deep one-class classifiers) and applications (e.g., outlier or anomaly detection), so that the results are less relevant for the purpose of this paper, which aims to provide a general and widely applicable framework for scenario decision making.

**Notation**  $\mathbb{N} = \{0, 1, 2, \ldots\}$ denotes the set of nonnegative integers. Given $m \in \mathbb{N}$, we let $[m] = \{1, \ldots, m\}$. Given a set $S$, $2^S$ (called the *power set* of $S$) denotes the set of all subsets of $S$, and $|S|$ denotes its cardinality. Given a probability measure $\mu$ and $m \in \mathbb{N}$, $\mu^m$ denotes its $m$-fold product, i.e., $\mu^m = \mu \otimes \cdots \otimes \mu$ ($m$ times). Given a set $S$, we let $S^* = \bigcup_{m=0}^{\infty} S^m$ be the set of all finite tuples with elements in $S$.

## 2 Problem statement

### 2.1 Risk-aware decision making and scenario decision making

Consider a set $\Xi$ of *decisions* (for instance, in Example 1, each $\xi \in \Xi$ represents a path in a configuration space; in Example 2, each $\xi \in \Xi$ represents an SVM classifier in a feature space). We consider *constraints* on our decisions (for instance, in Example 1, the path $\xi$ must avoid the obstacles; in Example 2, the classifier $\xi$ must classify the points correctly). Each constraint is represented by the set of decisions that satisfy it; hence, it is a subset of $\Xi$. We consider a set $X \subseteq 2^\Xi$ of constraints on $\Xi$. For each $x \in X$ and $\xi \in \Xi$, we say that $\xi$ *satisfies* the constraint $x$ if $\xi \in x$; otherwise $\xi$ *violates* $x$.

The goal of *risk-aware decision making* is to find a decision $\xi$ that satisfies the constraints in $X$ with a high probability. For that, we assume a probability measure $\mu$ on $X$.[1]

---
**Problem 1** (Risk-aware decision making)**.** For $\Xi$, $X$ and $\mu$ as above, and a given *risk tolerance* $\epsilon \in (0, 1]$, find $\xi \in \Xi$ such that $\mathsf{PSAT}_\mu(\xi) \doteq \mu(\{x \in X : \xi \in x\}) \geq 1 - \epsilon$.
---

Solving Problem 1 is challenging in general because the set $X$ can be very large or infinite, so that computing $\mathsf{PSAT}_\mu(\xi)$ for a given $\xi$ can already be challenging (Blackmore et al., 2010), let alone finding one that satisfies the risk tolerance. Furthermore, in many applications, the probability measure $\mu$ on $X$ is unknown. Indeed, it arises from complex phenomena—such as random noise, stochastic demand, random events, etc.—that are hard to capture or model accurately. In this context, sample-based methods such as *scenario decision making* provide a powerful and principled way to address these challenges. It consists in sampling $m$ constraints $x_1, \ldots, x_m$ from $X$ (independently with probability $\mu$) and making a decision based on these samples. Hence, a scenario decision algorithm maps tuples of constraints in $X$ to decisions in $\Xi$:

**Definition 1.** Let $\Xi$ be a decision space and $X$ a constraint space as above. A *scenario decision algorithm* is a function $A : X^* \to \Xi$, satisfying that for all $m \in \mathbb{N}$, $(x_1, \ldots, x_m) \in X^m$ and $i \in [m]$, $A(x_1, \ldots, x_m) \in x_i$.

*Remark* 1. The assumption made in Definition 1, called the *consistency assumption*, requires that the decision outputted by the algorithm satisfies the input samples.[2] This property is satisfied by a large class of scenario

---

[1] All subsets of $X$ or $X^*$ considered throughout the paper are supposed to be measurable with respect to the underlying distribution $\mu$. This includes the sets $\{x \in X : \xi \in x\}$ for all $\xi \in \Xi$ (used in Problem 1) and the sets $\{\boldsymbol{x} \in X^m : \mathsf{PSAT}_\mu(A(\boldsymbol{x})) \geq 1 - \epsilon\}$ for all $\epsilon > 0$ (used in equation 1). This is a common underlying assumption in scenario optimization (see, e.g., Garatti & Campi, 2021, footnote 6).

[2] We note that we do not assume the existence of a decision $\xi$ satisfying *all* $x \in X$. However, if a scenario decision algorithm exists for the problem, then for any *finite* set $\{x_1, \ldots, x_m\} \subseteq X$, there is a decision $\xi$ satisfying all $x_i$'s for $i \in [m]$.

decision algorithms in the literature (e.g., Campi & Garatti, 2008; Alamo et al., 2009; Margellos et al., 2014; Esfahani et al., 2015; Campi & Garatti, 2018; Campi et al., 2018; Garatti & Campi, 2022; Campi & Garatti, 2023; Rocchetta et al., 2024).

We illustrate the concept of scenario decision algorithm with two examples:

*Example* 1 (Path planning). Consider the problem of finding the shortest path between a source location $\mathsf{S}$ and a target location $\mathsf{T}$ while avoiding a randomly positioned obstacle, as depicted in Figure 1a. The path must belong to a set $\Xi$ of "admissible" paths (for instance, all paths that are smooth and have bounded curvature, or all paths that can be represented using $H$ via-points; Jankowski et al., 2023). Each possible position of the obstacle induces the constraint that the path must avoid the obstacle in that position. Since the position is random, this leads to the constraint that the path must avoid the obstacle with high probability.

More formally, let $\xi$ be a path and $c$ be the position of the center of the obstacle. Hence, the associated constraint on $\xi$, denoted by $\bar{x}(c)$, is $\xi \in \bar{x}(c)$ if and only if $\xi \cap B_\infty(c, \frac{1}{2}) = \emptyset$, where $B_\infty(c, r)$ is the $L^\infty$-ball centered at $c$ with radius $r$. We let $X = \{\bar{x}(c) : c \in \mathbb{R}^2\}$, and $\nu$ be a probability measure on the center $c$ of the obstacle. This induces naturally a probability measure $\mu$ on $X$, given by $\mu(S) = \nu(\{c \in \mathbb{R}^2 : \bar{x}(c) \in S\})$ (i.e., $\mu = \nu \circ \bar{x}^{-1}$).

When the probability measure $\nu$ is not known to the planner, the scenario approach can be used to circumvent this issue by requiring only sampled positions $c_1, \ldots, c_m$ of the center of the obstacle. For instance, a scenario decision algorithm can be defined by:

$$A(x_1, \ldots, x_m) = \arg\min_{\xi \in \Xi} \ \ell(\xi) \ \text{ s.t. } \ \xi \in x_i \ \ \forall i \in [m],$$

where $x_i = \bar{x}(c_i)$ for each $i \in [m]$ and $\ell(\xi)$ is the length of $\xi$.[3] A tie-breaking rule can be used if the minimizer is not unique. It is straightforward to see that $A$ satisfies Definition 1. ◁

*Example* 2 (SVM classifier). Consider a feature space $Z \subseteq \mathbb{R}^d$, and a probability measure $\nu$ on $Z \times \{-1, +1\}$. We seek an SVM classifier of the form $z \mapsto \text{sign}(\langle a, z \rangle + \beta)$, where $a$ and $\beta$ are parameters to determine, such that with high probability on $(z, \lambda) \in Z \times \{-1, +1\}$, $\lambda = \text{sign}(\langle a, z \rangle + \beta)$; see Figure 1b for an illustration.

For that, we let $\Xi$ be the set of decisions of the form $\xi = (a, \beta, \eta)$, where $a \in \mathbb{R}^d$, $\|a\| = 1$, $\beta \in \mathbb{R}$ and $\eta \in \mathbb{R}$. Furthermore, for each $(z, \lambda) \in Z \times \{-1, +1\}$, we define the constraint $\bar{x}(z, \lambda)$ by $(a, \beta, \eta) \in \bar{x}(z, \lambda)$ if and only if $\lambda(\langle a, z \rangle + \beta) \geq \eta$. The scenario decision algorithm is then defined as follows: given samples $(z_1, \lambda_1), \ldots, (z_m, \lambda_m)$ from $Z \times \{-1, +1\}$, we compute

$$A(x_1, \ldots, x_m) = \arg\max_{\xi = (a, \beta, \eta) \in \Xi} \ \eta \ \text{ s.t. } \ \xi \in x_i \ \ \forall i \in [m],$$

where $x_i = \bar{x}(z_i, \lambda_i)$ for each $i \in [m]$. A tie-breaking rule can be used if the maximizer is not unique. It is straightforward to see that $A$ satisfies Definition 1. ◁

This paper studies the correctness of scenario decision algorithms, in the sense that their output satisfies Problem 1 with high probability on the sampling of the input constraints. More precisely, we consider PAC (Probably Approximately Correct) guarantees, introduced by Valiant (1984) and standard in machine learning, capturing the fact that if sufficiently many i.i.d. samples of the constraints are used, then with high probability (with respect to the sampling of the constraints), the decision output by the algorithm has a risk below $\epsilon$ (i.e., satisfies Problem 1):

**Definition 2.** A scenario decision algorithm $A : X^* \to \Xi$ is *PAC* (*Probably Approximately Correct*) if for every $\delta \in (0, 1)$ and $\epsilon \in (0, 1)$, there is $N \in \mathbb{N}$ such that for all $m \in \mathbb{N} \cap [N, \infty)$ and all probability measure $\mu$ on $X$, it holds with probability at least $1 - \delta$ that if $m$ independent samples $x_1, \ldots, x_m$ are drawn from $\mu$, then $A(x_1, \ldots, x_m)$ satisfies Problem 1, i.e.,

$$\mu^m(\{\boldsymbol{x} \in X^m : \mathsf{PSAT}_\mu(A(\boldsymbol{x})) \geq 1 - \epsilon\}) \geq 1 - \delta, \tag{1}$$

---

[3]We assume that, given $(x_1, \ldots, x_m)$, there always exists a path satisfying $\xi \in x_i$ for all $i \in [m]$. This can be guaranteed for instance if we assume that the sampled positions of the obstacle never block completely the way from $S$ to $T$, or by having a "fallback" solution $\xi_{\text{fb}}$, which by definition satisfies all constraints $x \in X$, but has sufficiently large cost to not be preferred when there is a feasible path.

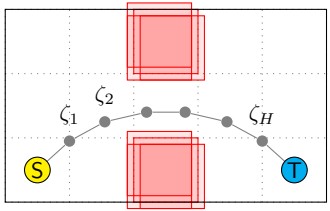
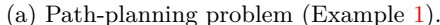

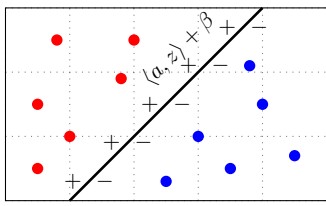

(a) Path-planning problem (Example 1).  (b) SVM classifier problem (Example 2).

Figure 1: (a) The path is parameterized by $H$ via-points (gray dots). The path must avoid the obstacles (red regions) with high probability, while having minimal length. (b) SVM classifier $\xi$ in the form of a separating hyperplane $\xi : z \mapsto \langle a, z \rangle + \beta$, where $z$ is a point in the feature space. The positive instances are represented in red, and the negative instances in blue.

where $\boldsymbol{x}$ is a shorthand for $(x_1, \ldots, x_m)$.

This leads to main problem considered in this paper:

**Problem 2** (PAC assessment)**.** Given a scenario decision algorithm $A : X^* \to \Xi$, decide if $A$ is PAC, and if it is, give sufficient values of $N$ as a function of $\delta$ and $\epsilon$.

## 2.2 Abstract scenario decision making

We introduce abstract scenario decision algorithms. The abstraction removes the explicit output decision "parametrization" and directly tracks which constraints are satisfied by the output. This simplifies the analysis of these algorithms.

To introduce the abstraction, let us start with a closer look at the definition of PAC algorithm (Definition 2). Given $\xi \in \Xi$, let $\mathsf{SAT}(\xi) = \{x \in X : \xi \in x\}$ be the set of constraints satisfied by $\xi$. The definition of $\mathsf{PSAT}$ in Problem 1 implies that $\mathsf{PSAT}_\mu(\xi) = \mu(\mathsf{SAT}(\xi))$. Hence, $\mathsf{PSAT}_\mu(A(\boldsymbol{x}))$ in equation 1 can be replaced by $\mu(A'(\boldsymbol{x}))$, where $A'(\boldsymbol{x}) = \mathsf{SAT}(A(\boldsymbol{x}))$. The algorithm $A' = \mathsf{SAT} \circ A$ is called the *abstraction* of $A$.

**Proposition 1.** *Consider a scenario decision algorithm $A : X^* \to \Xi$. The abstraction $A' = \mathsf{SAT} \circ A$ of $A$ satisfies (i) $A' : X^* \to 2^X$, (ii) for all $m \in \mathbb{N}$ and $(x_1, \ldots, x_m) \in X^m$, $\{x_1, \ldots, x_m\} \subseteq A'(x_1, \ldots, x_m)$, and (iii) for all $\boldsymbol{x} \in X^*$ and any probability measure $\mu$ on $X$, $\mathsf{PSAT}_\mu(A(\boldsymbol{x})) = \mu(A'(\boldsymbol{x}))$.*

*Proof.* (i) and (iii) are straightforward. (ii) is from the consistency of $A$. $\qquad\square$

This leads to the general definition of an abstract scenario decision algorithm:

**Definition 3.** Let $X$ be a constraint space. An *abstract scenario decision algorithm* is a function $A : X^* \to 2^X$, satisfying that for all $m \in \mathbb{N}$ and $(x_1, \ldots, x_m) \in X^m$, $\{x_1, \ldots, x_m\} \subseteq A(x_1, \ldots, x_m)$.

Definition 2 (PAC) translates directly to abstract scenario decision algorithms:

**Definition 4.** An abstract scenario decision algorithm $A : X^* \to 2^X$ is *PAC* (*Probably Approximately Correct*) if for every $\delta \in (0, 1)$ and $\epsilon \in (0, 1)$, there is $N \in \mathbb{N}$ such that for all $m \in \mathbb{N} \cap [N, \infty)$ and all probability measure $\mu$ on $X$, it holds with probability at least $1 - \delta$ that if $m$ independent samples $x_1, \ldots, x_m$ are drawn from $\mu$, then $A(x_1, \ldots, x_m)$ satisfies Problem 1, i.e.,

$$\mu^m(\{\boldsymbol{x} \in X^m : \mu(A(\boldsymbol{x})) \geq 1 - \epsilon\}) \geq 1 - \delta, \tag{2}$$

where $\boldsymbol{x}$ is a shorthand for $(x_1, \ldots, x_m)$.

In view of Proposition 1, it is sufficient to prove PAC results for abstract scenario decision algorithms, those will transfer directly to concrete scenario decision algorithms. Therefore, in the rest of this paper, we focus on abstract scenario decision algorithms, and call them simply "scenario decision algorithms".

*Example* 3 (Generated ideal). Let $X = \mathbb{N}$. Consider the (abstract) scenario decision algorithm $A : X^* \to 2^X$ defined by $A(x_1, \ldots, x_m) = \langle x_1, \ldots, x_m \rangle \doteq \{k_1 x_1 + \ldots + k_m x_m : k_1, \ldots, k_m \in \mathbb{Z}\} \cap \mathbb{N}$. It is easy to see that $A$ satisfies Definition 3. Indeed, for any $(x_1, \ldots, x_m) \in X^*$ and $i \in [m]$, it holds that $x_i \in \langle x_1, \ldots, x_m \rangle$ since $x_i = 0x_1 + \ldots + 0x_{i-1} + 1x_i + 0x_{i+1} + \ldots + 0x_m$. Modulo a straightforward identification between integers $x$ and their set of divisors $\mathsf{d}(x) = \{y \in \mathbb{N} : y \,|\, x\}$ (where $y \,|\, x \Leftrightarrow x \in y\mathbb{N}$), the algorithm $A$ can be obtained as the abstraction of a concrete algorithm $A_c$ with decision space $\Xi_c = \mathbb{N}$ and constraint set $X_c = \{\mathsf{d}(x) : x \in \mathbb{N}\}$, defined by $A_c(\mathsf{d}(x_1), \ldots, \mathsf{d}(x_m)) = \max\left(\bigcap_{i=1}^m \mathsf{d}(x_i)\right)$, i.e., $A_c(\mathsf{d}(x_1), \ldots, \mathsf{d}(x_m))$ is the greatest common divisor (gcd) of $\{x_1, \ldots, x_m\}$. It is known (see, e.g., Jacobson, 2009, p. 104) that $\xi$ is the gcd of $\{x_1, \ldots, x_m\}$ if and only if $\xi\mathbb{N} = \langle x_1, \ldots, x_m \rangle$, showing that $\mathsf{SAT}(A_c(\mathsf{d}(x_1), \ldots, \mathsf{d}(x_m))) = \{\mathsf{d}(x) : x \in \langle x_1, \ldots, x_m \rangle\}$. ◁

### 2.3 Stability and monotonicity

Before ending this section, we introduce two additional assumptions, called stability and monotonicity, on scenario decision algorithms, which will be used later in this paper when we discuss converse results.

The notion of stability is common in the scenario decision making (Campi & Garatti, 2008; Calafiore, 2010; Campi & Garatti, 2018; Garatti & Campi, 2022; 2021; Campi & Garatti, 2023; Rocchetta et al., 2024). It captures the property that if the returned decision satisfies some unsampled constraints, then adding these constraints to the tuple of sampled constraints does not change the output:

**Definition 5.** A scenario decision algorithm $A : X^* \to 2^X$ is *stable* if for all $\boldsymbol{x} \in X^*$ and $y \in A(\boldsymbol{x})$, it holds that $A(\boldsymbol{x}, y) = A(\boldsymbol{x})$.

The notion of monotonicity captures the property that the larger the set of sampled constraints, the larger the set of satisfied constraints:

**Definition 6.** A scenario decision algorithm $A : X^* \to 2^X$ is *monotone* if for all $(m, n) \in \mathbb{N}^2$, $(x_1, \ldots, x_m) \in X^m$ and $(y_1, \ldots, y_n) \in X^n$, $\{x_1, \ldots, x_m\} \subseteq \{y_1, \ldots, y_n\}$ implies $A(x_1, \ldots, x_m) \subseteq A(y_1, \ldots, y_n)$.

*Example* 4. The scenario decision algorithm $A$ in Example 3 is stable and monotone. Indeed: 1) For the stability: if $y \in \langle x_1, \ldots, x_m \rangle$, then $y = \sum_{i=1}^m \ell_i x_i$ for $\ell_i \in \mathbb{Z}$, $i \in [m]$, so that $\langle x_1, \ldots, x_m, y \rangle = \langle x_1, \ldots, x_m \rangle$. 2) For the monotonicity: it is clear that $\{x_1, \ldots, x_m\} \subseteq \{y_1, \ldots, y_n\}$ implies $\langle x_1, \ldots, x_m \rangle \subseteq \langle y_1, \ldots, y_n \rangle$. ◁

*Example* 5. The scenario decision algorithms searching for the smallest enclosing rectangle or polytope (as in, e.g., Eppstein & Erickson, 1994; Chan & Har-Peled, 2021 with no outliers) are stable and monotone.

## 3 The internal growth

In this section, we present the main concept introduced in this paper: the *internal growth*. This concept will be leveraged in the next section to derive a PAC sufficient condition for scenario decision algorithms.

The notion of internal growth is based on the notion of internal projected hypothesis class, which we introduce first. Both notions are extensions of the notions of growth function and projected hypothesis class in binary-classification learning (see, e.g., Shalev-Shwartz & Ben-David, 2014; Mohri et al., 2018) to scenario decision algorithms; see also Remark 2 below.

**Definition 7.** Let $A$ be a scenario decision algorithm and $S \subseteq X$. The *internal projected hypothesis class* of $S$ with respect to $A$ is the set

$$\sigma(S; A) = \{A(\boldsymbol{x}) \cap S : \boldsymbol{x} \in S^*\}.$$

The *internal projected hypothesis size* of $S$ with respect to $A$ is defined by $\rho(S; A) = |\sigma(S; A)|$.

*Remark* 2. The notion of internal projected hypothesis class contrasts with the classical notion of projected hypothesis class in binary-classification learning, which corresponds to the set $\{A(\boldsymbol{x}) \cap S : \boldsymbol{x} \in X^*\}$; see also Definition 10, later in the paper.

*Example* 6. Let $X$ and $A$ be as in Example 3. Let $S = \{2, 5, 6\}$. Then, one can check that

$$
\begin{array}{llll}
A(\varepsilon) \cap S = \emptyset, & A(5) \cap S = \{5\}, & A(2, 5) \cap S = S, & A(5, 6) \cap S = S, \\
A(2) \cap S = \{2, 6\}, & A(6) \cap S = \{6\}, & A(2, 6) \cap S = \{2, 6\}, & A(2, 5, 6) \cap S = S,
\end{array}
$$

where $\varepsilon$ is the empty tuple. Since $A$ is monotone, input tuples with the same set of elements give the same output (e.g., $A(2,5,5) = A(5,2)$). Hence, $\sigma(S; A) = \{\emptyset, \{5\}, \{6\}, \{2,6\}, S\}$ and $\rho(S; A) = 5$. Note that in this example, the internal projected hypothesis class coincides with the projected hypothesis class. ◁

Finally, the internal growth function gives the maximal value of the internal projected hypothesis size for all sets of a given cardinality:

**Definition 8.** Let $A : X^* \to 2^X$ be a scenario decision algorithm. The *internal growth function* $\tau(\cdot; A) : \mathbb{N} \to \mathbb{N}$ of $A$ is defined by

$$\tau(m; A) = \max\left\{\rho(S; A) : S \subseteq X, |S| \leq m\right\}.$$

Clearly, $\tau(m; A) \leq 2^m$. We say that $A$ has slow internal growth if $\tau$ grows at a sub-exponential rate:

**Definition 9.** Let $A : X^* \to 2^X$ be a scenario decision algorithm. We say that $A$ has *slow internal growth* if for all $r > 1$, it holds that $\tau(m; A) \in O(r^m)$, i.e., there is $C \geq 0$ such that for all $m \in \mathbb{N}$, $\tau(m; A) \leq Cr^m$.

In the following, we drop the reference to $A$ in $\sigma$, $\rho$ and $\tau$, when $A$ is clear from the context.

*Example* 7. Let $X$ and $A$ be as in Example 3. It holds that $\tau(m) = 2^m$. Indeed, let $m \in \mathbb{N}$ and $\mathcal{P}$ be the set containing the first $m$ prime numbers. Consider the set $S = \{P/q : q \in \mathcal{P}\}$ where $P = \prod_{p \in \mathcal{P}} p$. It is not difficult to show that for any $r \in \mathbb{N}$ and $(x_1, \ldots, x_r) \in S^r$, $A(x_1, \ldots, x_r) \cap S = \{x_1, \ldots, x_r\}$.[4] Hence, $\sigma(S) = 2^S$ and $\rho(S) = 2^{|S|} = 2^m$, which implies that $A$ has *not* slow internal growth. ◁

## 4 Slow internal growth implies PAC

In this section, we present the main result of the paper, stating that the slow internal growth property implies that the algorithm is PAC, and we derive sufficient bounds on $N$ (cf. Problem 2) based on the internal growth function of $A$:

**Theorem 1.** *Let $A : X^* \to 2^X$ be a scenario decision algorithm. If $A$ has slow internal growth, then $A$ is PAC. Furthermore, $N$ in Definition 2 can be any $N$ such that (i) $\epsilon N \geq 8$, and (ii) for all $m \in \mathbb{N} \cap [N, \infty)$, $2e^{-\epsilon m/4}\tau(2m) \leq \delta$. We note that $N \in \mathbb{N}$ satisfying (i) and (ii) always exists since, by Definition 9, for all $\epsilon > 0$, there exists $C \geq 0$ such that for all $m \in \mathbb{N}$, $\tau(2m) \leq Ce^{\epsilon m/5}$.*

**Proof of Theorem 1**

We will prove the theorem in several steps. The proof is inspired from the proof of the frequency of $\epsilon$-nets by Haussler & Welzl (1987), but adapts to include internal projected hypothesis classes.

Fix $\epsilon \in (0, 1)$ and let $m \in \mathbb{N}$ be such that $\epsilon m \geq 8$. Fix a probability measure $\mu$ on $X$ and consider the following three sets:

$$B = \{(x_1, \ldots, x_m) \in X^m : \mu(A(x_1, \ldots, x_m)) < 1 - \epsilon\},$$

$$\tilde{B}_1 = \left\{(x_1, \ldots, x_{2m}) \in X^{2m} : \mu(A(x_1, \ldots, x_m)) < 1 - \epsilon, |\{x_{m+1}, \ldots, x_{2m}\} \setminus A(x_1, \ldots, x_m)| > \frac{\epsilon m}{2}\right\},$$

$$\tilde{B}_2 = \left\{(x_1, \ldots, x_{2m}) \in X^{2m} : \exists T \in \sigma(\{x_1, \ldots, x_{2m}\}), \{x_1, \ldots, x_m\} \subseteq T, |\{x_{m+1}, \ldots, x_{2m}\} \setminus T| > \frac{\epsilon m}{2}\right\}.$$

**Lemma 1.** *Let $A$ be as in Theorem 1, and $\tilde{B}_1$ and $\tilde{B}_2$ be as above. It holds that $\tilde{B}_1 \subseteq \tilde{B}_2$.*

*Proof.* Straightforward since $T$ can be chosen equal to $A(x_1, \ldots, x_m)$ in the definition of $\tilde{B}_2$. □

The following lemma is instrumental, and inspired by Haussler & Welzl (1987, Lemma 3.4):

**Lemma 2.** *Let $A$ be as in Theorem 1, and $B$ and $\tilde{B}_1$ be as above. It holds that $\mu^m(B) \leq 2\mu^{2m}(\tilde{B}_1)$.*

---

[4]Indeed, $A(x_1, \ldots, x_r) = g\mathbb{N}$ where $g$ is the gcd of $\{x_1, \ldots, x_r\}$, which satisfies $g = P/(q_1 \cdots q_s)$ where $q_1, \ldots, q_s \in \mathcal{P}$ are distinct and such that $\{x_1, \ldots, x_r\} = \{P/q_1, \ldots, P/q_s\}$. If $y \in S \setminus \{x_1, \ldots, x_r\}$, then $y = P/q$ for some $q \in \mathcal{P} \setminus \{q_1, \ldots, q_s\}$, so that $g \in q\mathbb{N}$, which implies that $y \notin g\mathbb{N}$. To give an example, take $m = 3$, so that $\mathcal{P} = \{2, 3, 5\}$, $P = 30$, and $S = \{15, 10, 6\}$. For $\boldsymbol{x} = (15, 10)$, we get $A(\boldsymbol{x}) = 5\mathbb{N}$ so that $A(\boldsymbol{x}) \cap S = \{15, 10\}$; for $\boldsymbol{x} = (15, 6)$, we get $A(\boldsymbol{x}) = 3\mathbb{N}$ so that $A(\boldsymbol{x}) \cap S = \{15, 6\}$; for $\boldsymbol{x} = (10, 6)$, we get $A(\boldsymbol{x}) = 2\mathbb{N}$ so that $A(\boldsymbol{x}) \cap S = \{10, 6\}$; etc.

*Proof.* To simplify notation, we denote $\boldsymbol{x}_1 \doteq (x_1, \ldots, x_m)$ and $\boldsymbol{x}_2 \doteq (x_{m+1}, \ldots, x_{2m})$. Note that $\mu^{2m}(\tilde{B}_1) = \mathbb{E}_{\boldsymbol{x}_1}\mathbb{E}_{\boldsymbol{x}_2}\mathbf{1}[(\boldsymbol{x}_1, \boldsymbol{x}_2) \in \tilde{B}_1]$. Since $(\boldsymbol{x}_1, \boldsymbol{x}_2) \in \tilde{B}_1$ implies $\boldsymbol{x}_1 \in B$, it follows that

$$\mu^{2m}(\tilde{B}_1) = \mathbb{E}_{\boldsymbol{x}_1}\big(\mathbf{1}[\boldsymbol{x}_1 \in B] \cdot \mathbb{E}_{\boldsymbol{x}_2}\mathbf{1}[(\boldsymbol{x}_1, \boldsymbol{x}_2) \in \tilde{B}_1]\big).$$

Now, fix $\boldsymbol{x}_1 \in B$ and let $\rho = 1 - \mu(A(\boldsymbol{x}_1)) > \epsilon$. It holds that

$$\mathbb{E}_{\boldsymbol{x}_2}\mathbf{1}[(\boldsymbol{x}_1, \boldsymbol{x}_2) \notin \tilde{B}_1] = \sum_{k=0}^{\lfloor \frac{\epsilon m}{2} \rfloor} \binom{m}{k} \rho^k (1-\rho)^{m-k}$$

since it is the probability that at most $\lfloor \frac{\epsilon m}{2} \rfloor$ elements of $\boldsymbol{x}_2$ belong to $X \setminus A(\boldsymbol{x}_1)$, where the elements are sampled i.i.d. and the marginal probability of any element to belong to $X \setminus A(\boldsymbol{x}_1)$ is $\rho$.[5] Since $\lfloor \frac{\epsilon m}{2} \rfloor < \frac{\rho m}{2}$, and by Chebyshev's inequality, we obtain that

$$\mathbb{E}_{\boldsymbol{x}_2}\mathbf{1}[(\boldsymbol{x}_1, \boldsymbol{x}_2) \notin \tilde{B}_1] \leq \frac{m\rho(1-\rho)}{(\rho m/2)^2} \leq \frac{4}{\rho m} \leq \frac{1}{2},$$

where the last inequality follows from the assumption that $\epsilon m \geq 8$. Hence, $\mathbb{E}_{\boldsymbol{x}_2}\mathbf{1}[(\boldsymbol{x}_1, \boldsymbol{x}_2) \in \tilde{B}_1] \geq \frac{1}{2}$ and

$$\mu^{2m}(\tilde{B}_1) \geq \frac{1}{2}\mathbb{E}_{\boldsymbol{x}_1}\mathbf{1}[\boldsymbol{x}_1 \in B\} = \frac{1}{2}\mu^m(B),$$

concluding the proof. $\qquad\square$

**Lemma 3.** *It holds that $\mu^{2m}(\tilde{B}_2) \leq \left(1 - \frac{\epsilon}{4}\right)^m \tau(2m)$.*

*Proof.* Let $\alpha = \frac{\epsilon m}{2}$. We let $\mathcal{J}$ be the uniform distribution on the subsets of $m$ elements from the set $[2m]$. Note that

$$\mu^{2m}(\tilde{B}_2) = \mathbb{E}_{(x_1, \ldots, x_{2m})} \max_{T \in \sigma(\{x_1, \ldots, x_{2m}\})} \left(\mathbf{1}[\{x_1, \ldots, x_m\} \subseteq T] \cdot \mathbf{1}[|\{x_{m+1}, \ldots, x_{2m}\} \setminus T| > \alpha]\right)$$

$$= \mathbb{E}_{(x_1, \ldots, x_{2m})} \max_{T \in \sigma(\{x_1, \ldots, x_{2m}\})} \left(\mathbf{1}[\{x_1, \ldots, x_m\} \subseteq T] \cdot \mathbf{1}[|\{x_1, \ldots, x_{2m}\} \setminus T| > \alpha]\right)$$

$$= \mathbb{E}_{(x_1, \ldots, x_{2m})}\mathbb{E}_{\{j_1, \ldots, j_m\} \sim \mathcal{J}} \max_{T \in \sigma(\{x_1, \ldots, x_{2m}\})} \left(\mathbf{1}[\{x_{j_1}, \ldots, x_{j_m}\} \subseteq T] \cdot \mathbf{1}[|\{x_1, \ldots, x_{2m}\} \setminus T| > \alpha]\right)$$

(since one can replace $\{x_1, \ldots, x_m\}$ by $\{x_{j_1}, \ldots, x_{j_m}\}$)

$$\leq \mathbb{E}_{(x_1, \ldots, x_{2m})}\mathbb{E}_{\{j_1, \ldots, j_m\} \sim \mathcal{J}} \sum_{T \in \sigma(\{x_1, \ldots, x_{2m}\})} \left(\mathbf{1}[\{x_{j_1}, \ldots, x_{j_m}\} \subseteq T] \cdot \mathbf{1}[|\{x_1, \ldots, x_{2m}\} \setminus T| > \alpha]\right)$$

(because the sum is larger than or equal to the max)

$$= \mathbb{E}_{(x_1, \ldots, x_{2m})} \sum_{T \in \sigma(\{x_1, \ldots, x_{2m}\})} \left(\mathbf{1}[|\{x_1, \ldots, x_{2m}\} \setminus T| > \alpha] \cdot \mathbb{E}_{\{j_1, \ldots, j_m\} \sim \mathcal{J}}\mathbf{1}[\{x_{j_1}, \ldots, x_{j_m}\} \subseteq T]\right)$$

$$\leq \mathbb{E}_{(x_1, \ldots, x_{2m})} \sum_{T \in \sigma(\{x_1, \ldots, x_{2m}\})} \left(\mathbf{1}[|\{x_1, \ldots, x_{2m}\} \setminus T| > \alpha] \left(1 - \frac{\alpha}{2m}\right)^m\right)$$

(since the probability of choosing $m$ elements in $\{x_1, \ldots, x_{2m}\}$ that are in $T$ is at most $(1 - c/2m)^m$, where $c = |\{x_1, \ldots, x_{2m}\} \setminus T| > \alpha$)

$$\leq \mathbb{E}_{(x_1, \ldots, x_{2m})} \left(\rho(\{x_1, \ldots, x_{2m}\}) \left(1 - \frac{\alpha}{2m}\right)^m\right) \leq \left(1 - \frac{\epsilon}{4}\right)^m \tau(2m).$$

This concludes the proof. $\qquad\square$

---

[5]Hence, the number of elements of $\boldsymbol{x}_2$ that belong to $X \setminus A(\boldsymbol{x}_1)$ follows a Binomial distribution with parameters $m$ (number of trials) and $\rho$ (probability of success of each trial). Its mean is given by $\mu = m\rho$ and its variance by $\sigma^2 = m\rho(1-\rho)$.

*Proof of Theorem 1.* Fix $\epsilon \in (0, 1)$ and $\delta \in (0, 1)$. Let $m \in \mathbb{N}$ be such that $2 \left(1 - \frac{\epsilon}{4}\right)^m \tau(2m) \leq \delta$ and $\epsilon m \geq 8$. By Lemmas 1, 2 and 3, we have that

$$\mu^m(B) \leq 2\mu^{2m}(\tilde{B}_1) \leq 2\mu^{2m}(\tilde{B}_2) \leq 2e^{-\epsilon m/4}\tau(2m) \leq \delta.$$

Hence,

$$\text{LHS of equation } 2 = \mu^m(X^m \setminus B) = 1 - \mu^m(B) \geq 1 - \delta.$$

To conclude the proof, note that $2e^{-\epsilon m/4}\tau(2m) \leq \delta$ implies $2 \left(1 - \frac{\epsilon}{4}\right)^m \tau(2m) \leq \delta$ since $1 + x \leq e^x$. $\qquad\square$

## 5 Connections with other PAC criteria

In this section, we discuss the connection of the slow internal growth property with other PAC sufficient conditions available in the literature, namely the finite VC dimension criterion (Alamo et al., 2009) and the existence of a compression scheme (Margellos et al., 2015).[6] We show that both criteria imply that the internal growth function is polynomial, thereby implying the slow internal growth property. This shows that the latter is at least as general as the union of both existing criteria. The examples in Section 7 will show that it is in fact *strictly* more general (in particular, the scenario decision algorithm in Example 9 features super-polynomial, yet sub-exponential, internal growth). Next to this, we also introduce a novel quantity, called the $k$-VC dimension, which refines the notion of VC dimension, and we show that finiteness of this quantity for all $k$ is necessary and sufficient for the slow internal growth property.

### 5.1 Finite VC dimension and $k$-VC dimension

We first present the PAC sufficient condition from Alamo et al. (2009), based on the finiteness of the VC dimension, and show that our sufficient condition is more general. To present this condition, we need to first recall the notions of projected hypothesis class, growth function and VC dimension, introduced by Vapnik & Chervonenkis (1971) in the context of binary classification, and adapted to scenario decision algorithms by Alamo et al. (2009):

**Definition 10.** Let $A : X^* \to 2^X$ be a scenario decision algorithm and let $S \subseteq X$. The *projected hypothesis class* of $S$ with respect to $A$ is the set

$$\sigma'(S; A) = \{A(\boldsymbol{x}) \cap S : \boldsymbol{x} \in X^*\}.$$

The *projected hypothesis size* of $S$ with respect to $A$ is defined by $\rho'(S; A) = |\sigma'(S; A)|$.
The *growth function* $\tau'(\,\cdot\,; A) : \mathbb{N} \to \mathbb{N}$ of $A$ is defined by

$$\tau'(m; A) = \max \{\rho'(S; A) : S \subseteq X, |S| \leq m\}.^{7}$$

The *VC dimension* of $A$ is the supremum of all $m \in \mathbb{N}$ such that $\tau'(m; A) = 2^m$.

It is shown by Alamo et al. (2009) that if $A$ has finite VC dimension, then $A$ is PAC. In fact, if $A$ has finite VC dimension, it also holds that $A$ has slow internal growth, thereby showing that the slow internal growth property is more general than the finite VC dimension property:

**Proposition 2.** *Let $A : X^* \to 2^X$ be a scenario decision algorithm. If $A$ has finite VC dimension, then $A$ has slow internal growth. In particular, it holds that $\tau(m) \leq \max \left\{2^d, \left(\frac{em}{d}\right)^d\right\}$.*

*Proof.* It is straightforward to see that for all $S \subseteq X$, $\sigma(S) \subseteq \sigma'(S)$. Hence, $\rho(S) \leq \rho'(S)$ and $\tau(m) \leq \tau'(m)$. Let $d < \infty$ be the VC dimension of $A$. By Sauer–Shelah's lemma (see, e.g., Shalev-Shwartz & Ben-David, 2014, Lemma 6.10), it holds that for all $m \in \mathbb{N} \cap (d, \infty)$, $\tau'(m) \leq \left(\frac{em}{d}\right)^d$. Also, when $m \in \mathbb{N} \cap [0, d]$, it holds that $\tau'(m) \leq 2^d$, concluding the proof. $\qquad\square$

---

[6] See also Rocchetta et al. (2024) for a survey.
[7] Clearly, $\tau'(m; A) \leq 2^m$.

Next, we introduce a novel quantity inspired from the VC dimension, which we call the $k$-VC dimension and is always smaller than the VC dimension. We then show an equivalence result with the slow internal growth property. The purpose of this equivalent criterion is to ease the verification of the slow growth property. The $k$-VC dimension is defined as follows:

**Definition 11.** Let $A : X^* \to 2^X$ be a scenario decision algorithm and $k \in \mathbb{N}$. The *$k$-VC dimension* of $A$ is the supremum of all $m \in \mathbb{N}$ for which there are sets $S \subseteq X$ and $R \subseteq X$ such that $|S| = m$, $|R| \leq km$ and $\{A(\boldsymbol{x}) \cap S : \boldsymbol{x} \in R^*\} = 2^S$.

We show that the $k$-VC dimension of a scenario decision algorithm is finite for all $k \in \mathbb{N}$ if and only if the algorithm has slow internal growth:

**Theorem 2.** *Let $A : X^* \to 2^X$ be a scenario decision algorithm. The following propositions are equivalent: (i) $A$ has finite $k$-VC dimension for all $k \in \mathbb{N}$; (ii) $A$ has slow internal growth.*

*Proof.* 1) Proof of (ii) $\Rightarrow$ (i): For a proof by contraposition, let $k \in \mathbb{N}$ and assume that $A$ has infinite $k$-VC dimension. We will show that for infinitely many $m \in \mathbb{N}$, $\tau(m) \geq 2^{m/(k+1)}$, which will conclude the proof.

For that, fix $N \in \mathbb{N}$. Let $m \in \mathbb{N} \cap [N, \infty)$ such that there are $S$ and $R$ as in Definition 11. Let $U = S \cup R$. It holds that $|U| \leq (k+1)m$, and that $2^S \subseteq \sigma(U)$. Hence, $\tau((k+1)m) \geq 2^m$, concluding the proof.

2) Proof of (i) $\Rightarrow$ (ii): Fix $r > 1$. Let $k \in \mathbb{N}$ such that $(ek) \leq r^k$. Let $d \in \mathbb{N}$ be the $k$-VC dimension of $A$. We show that for all $m \in \mathbb{N} \cap [kd, \infty)$, $\tau(m) \leq r^m$, which will conclude the proof of 2).

For that, fix $m \in \mathbb{N} \cap [kd, \infty)$. Fix $S \subseteq X$ with $|S| \leq m$. Let $U \subseteq S$ be such that $\{A(\boldsymbol{x}) \cap U : \boldsymbol{x} \in S^*\} = 2^U$. It holds that $|U| \leq m/k$: indeed, if $|U| > m/k$, then $|S| \leq k|U|$, and since the $k$-VC dimension of $A$ is $d$, it follows that $|U| \leq d$, a contradiction. Since $U$ was arbitrary, it follows that the largest cardinality $c$ of a set $U \subseteq S$ such that $\{A(\boldsymbol{x}) \cap U : \boldsymbol{x} \in S^*\} = 2^U$ is at most $m/k$. Hence, from the Sauer–Shelah lemma (see, e.g., Shalev-Shwartz & Ben-David, 2014, Lemma 6.10), it holds that

$$\rho(S) \leq \left(\frac{em}{c}\right)^c \leq (ek)^{m/k} \leq r^m,$$

where the last inequality follows from the definition of $k$. $\qquad\square$

## 5.2 Compression scheme

Finally, we present the PAC sufficient condition from Margellos et al. (2015), which is based on the notion of compression, first introduced by Littlestone & Warmuth (1986) in the context of binary classification. We then show that our PAC sufficient condition is more general.

**Definition 12.** Let $A : X^* \to 2^X$ be a scenario decision algorithm and $d \in \mathbb{N}$. A *compression scheme* of size $d$ for $A$ is a pair of functions $(A_c, A_r)$,[8] wherein $A_c : X^* \to X^d$ and $A_r : X^d \to 2^X$ satisfy that for all $m \in \mathbb{N}$ and $(x_1, \ldots, x_m) \in X^m$, $A_c(x_1, \ldots, x_m) \in \{x_1, \ldots, x_m\}^d$ and $A(x_1, \ldots, x_m) = A_r(A_c(x_1, \ldots, x_m))$.

It is shown by Margellos et al. (2015) that if $A$ admits a compression scheme, then $A$ is PAC. In fact, we can show that if $A$ admits a compression scheme, then $A$ has slow internal growth, thereby showing that the slow internal growth property is more general than the compression property:

**Proposition 3.** *Let $A : X^* \to 2^X$ be a scenario decision algorithm. If $A$ admits a compression scheme, then $A$ has slow internal growth. In particular, it holds that $\tau(m) \leq m^d$.*

*Proof.* Let $(A_c, A_r)$ be a compression scheme of size $d$ for $A$. Let $m \in \mathbb{N}$, and let $S \subseteq X$ such that $|S| \leq m$. For each $\boldsymbol{x} \in S^*$, it holds that $A_c(\boldsymbol{x}) \subseteq S^d$. Hence, $\sigma(S) \subseteq \{A_r(\boldsymbol{y}) : \boldsymbol{y} \in S^d\}$. Thus, $\rho(S) \leq m^d$. This shows that $\tau(m) \leq m^d$, concluding the proof. $\qquad\square$

*Remark* 3. Propositions 2 and 3 show that when the scenario decision algorithm has finite VC dimension or admits a compression scheme, then its internal growth function is polynomial. In Section 7.2, we study a scenario decision algorithm whose internal growth function is super-polynomial. However, since it is still sub-exponential, we can prove that the algorithm is PAC by using the slow internal growth property.

---

[8]$A_c$ is called the *compression map* and $A_r$ the *reconstruction map*.

## 6 When is the sufficient condition necessary?

In this last theoretical section, we investigate the question of "when is the slow internal growth property not only sufficient, but also necessary to be PAC"? We show a converse result for the case of stable monotone algorithms (cf. Section 2.3):

**Theorem 3.** *Let $A : X^* \to 2^X$ be a stable monotone scenario decision algorithm. If $A$ has not slow internal growth, then $A$ is not PAC.*

The proof relies on the following result from Berger & Jungers (2025, Section 4):

**Lemma 4** (Berger & Jungers, 2025, Theorem 4)**.** *Let $A : X^* \to 2^X$ be a scenario decision algorithm. If for all $d \in \mathbb{N}$, there is $m \in \mathbb{N} \cap [d, \infty)$ such that $\tau(m) = 2^m$, then $A$ is not PAC.*

*Remark* 4. In Berger & Jungers (2025, Section 4), the largest $m \in \mathbb{N}$ such that $\tau(m) = 2^m$ is called the *dVC dimension* of $A$. Note that in Lemma 4, we do not require that $A$ is stable or monotone.

The proof of Theorem 3 is concluded by showing that the dVC dimension of $A$ (cf. Remark 4) is infinite if $A$ has not slow internal growth:

**Lemma 5.** *Let $A : X^* \to 2^X$ be a stable monotone scenario decision algorithm. If $A$ has not slow internal growth, then for all $d \in \mathbb{N}$, there is $m \in \mathbb{N} \cap [d, \infty)$ such that $\tau(m) = 2^m$.*

*Proof.* Let $d \in \mathbb{N}$. Assume that $A$ has not slow internal growth. By Proposition 2, it follows that $A$ has infinite VC dimension. Hence, there exist $m \in \mathbb{N} \cap [d, \infty)$ and $S \subseteq X$ such that $|S| = m$ and $\sigma'(S) = 2^S$. Using the assumptions of stability and monotonicity, we will show that $\sigma(S) = 2^S$, so that $\rho(S) = 2^m$ and $\tau(m) = 2^m$. Therefore, let $n \in \mathbb{N}$ and $T \doteq \{x_1, \ldots, x_n\} \subseteq S$. We will show that $A(x_1, \ldots, x_n) \cap S = T$. The definition of $S$ implies that there is $\boldsymbol{x} \in X^*$ such that $A(\boldsymbol{x}) \cap S = T$. Fix such a $\boldsymbol{x}$. From the stability, it holds that $A(\boldsymbol{x}, x_1, \ldots, x_n) \cap S = A(\boldsymbol{x})$. From the monotonicity, it follows that $A(x_1, \ldots, x_n) \cap S \subseteq T$. From the consistency, it holds that $T \subseteq A(x_1, \ldots, x_n)$. Hence, $A(x_1, \ldots, x_n) \cap S = T$. Since $n \in \mathbb{N}$ and $T \subseteq S$ were arbitrary, this implies that $\sigma(S) = 2^S$, concluding the proof. $\qquad\square$

*Proof of Theorem 3.* Combine Lemmas 4 and 5. $\qquad\square$

## 7 Illustrative examples

We present three examples illustrating our results. We first present a simple example, showing elementarily that the slow internal growth property is strictly more general than the finite VC dimension condition and the existence of a compression scheme (cf. Section 5). Then, we present a more involved example, showing that the slow internal growth property can analyze non-trivial scenario optimization problems, which are also not tractable with the other criteria. Finally, we revisit Example 1 and show that the slow internal growth property provides PAC guarantees for the associated path-planning algorithm.

### 7.1 Simple example

This example shows elementarily that the slow internal growth property is strictly more general than the finite VC dimension condition and the existence of a compression scheme. The example relies on the existence of scenario decision algorithms satisfying the finite VC dimension condition, but not the existence of a compression scheme, and vice-versa, as shown by Berger & Jungers (2025). It goes as follows:

*Example* 8 (Union of scenario decision algorithms). Let $X_1$ be a constraint set and $A_1 : X_1^* \to 2^{X_1}$ be a scenario decision algorithm that has finite VC dimension, but does not admit a compression scheme (for instance, take the algorithm in Berger & Jungers, 2025, Section III.B). Similarly, let $X_2$ be a constraint set and $A_2 : X_2^* \to 2^{X_2}$ be a scenario decision algorithm that admits a compression scheme, but has infinite VC dimension (for instance, take the algorithm in Berger & Jungers, 2025, Section III.A). Without loss of generality, assume that $X_1 \cap X_2 = \emptyset$.

Let $X = X_1 \cup X_2$, and define $A : X^* \to 2^X$ by $A(x_1, \ldots, x_m) = A_1(x_{i_1}, \ldots, x_{i_n}) \cup A_2(x_{j_1}, \ldots, x_{j_{m-n}})$, where $\{x_{i_1}, \ldots, x_{i_n}\} \subseteq X_1$, $\{x_{j_1}, \ldots, x_{j_{m-n}}\} \subseteq X_2$, $i_1 < \ldots < i_n$ and $j_1 < \ldots < j_{m-n}$. In other words, the input

tuple of sampled constraints $x_1, \ldots, x_m$ is divided into two tuples $x_{i_1}, \ldots, x_{i_n}$ and $x_{j_1}, \ldots, x_{j_{m-n}}$ depending on whether the sampled constraint belongs to $X_1$ or $X_2$, and then passed respectively to $A_1$ and $A_2$. Once this is done, the set returned by $A_1$ is combined with the one by $A_2$ to give the final output of $A$. ◁

**Proposition 4.** *The scenario decision algorithm $A$ in Example 8 has infinite VC dimension and does not admit a compression scheme.*

*Proof.* Straightforward from the assumptions on $A_1$ and $A_2$, and since for each $s \in \{1, 2\}$ and all $\boldsymbol{x} \in X_s^*$, $A(\boldsymbol{x}) = A_s(\boldsymbol{x})$. □

We show that $A$ has slow internal growth. This comes from a more general fact that the *union* of finitely many scenario decision algorithms preserves the slow internal growth property:

**Proposition 5.** *Let $p \in \mathbb{N}_{>0}$. For each $s \in [p]$, let $X_s$ be a constraint set and $A_s : X_s^* \to 2^{X_s}$ be a scenario decision algorithm. Define $X = \bigcup_{s=1}^p X_s$ and $A : X^* \to 2^X$ by $A(\boldsymbol{x}) = \bigcup_{s=1}^p A_s(\boldsymbol{x}_s)$, where for each $s \in [p]$, $\boldsymbol{x}_s$ is the subtuple of $\boldsymbol{x}$ containing the elements of $\boldsymbol{x}$ that are in $X_s$ and only those ones.[9] It holds that for all $m \in \mathbb{N}$, $\tau(m; A) \leq \prod_{s=1}^p \tau(m; A_s)$. Hence, if for each $s \in [p]$, $A_s$ has slow internal growth, then $A$ has slow internal growth.*

*Proof.* To prove the bound on the internal growth function, let $S \subseteq X$. For each $s \in [p]$, let $S_s = S \cap X_s$. It holds that $\sigma(S) = \{\bigcup_{s=1}^p (A_s(\boldsymbol{x}_s) \cap X_s) : \boldsymbol{x}_s \subseteq S_s^* \; \forall s \in [p]\}$. Hence, $\rho(S; A) \leq \prod_{s=1}^p \rho(S_s; A_s)$. Thus, for all $m \in \mathbb{N}$, $\tau(m; A) \leq \prod_{s=1}^p \tau(m; A_s)$, concluding the proof of the bound on the internal growth function. The proof of the preservation of the slow internal growth property is straightforward. □

**Corollary 1.** *The scenario decision algorithm $A$ in Example 8 has slow internal growth.*

*Proof.* Straightforward from Proposition 5 and the fact that $A_1$ and $A_2$ have slow internal growth (consequence of Propositions 2 and 3, respectively). □

This concludes the first example, showing elementarily that the slow internal growth property is strictly more general than the other criteria discussed in Section 5.

## 7.2 Scenario optimization problem

In this section, we present a more involved, and concrete, example of scenario decision algorithm, that cannot be proved PAC using the criteria discussed in Section 5, but can be proved so using the slow internal growth criterion. Furthermore, this example exhibits a clear distinction with the algorithms that can be studied with the criteria in Section 5 (or a combination of them using Proposition 5) since its internal growth function is super-polynomial (it grows in $2^{\sqrt{m}}$). The example can be seen as a two-stage optimization problem, where one first optimizes an objective with respect to a variable $u$, and then optimizes a second objective with respect to a second variable $v$, with constraints given by the optimizer $u^\star$ of the first stage. It goes as follows:

*Example* 9 (Two-stage scenario optimization). Let $u_1, u_2$ be decision variables with values in $\mathbb{N}$, and $v_0, v_1, \ldots$ be decision variables with values in $\{0, 1\}$. Denote $u = (u_1, u_2)$ and $v = (v_0, v_1, \ldots)$. Let $\Xi = \mathbb{N}^2 \times \{0, 1\}^{\mathbb{N}}$ be the set of values of $(u, v)$. Let $X_1$ be a set of constraints of the form $X_1 = \{\bar{x}_1(c) : c \in \mathbb{N}^2\}$, where $\bar{x}_1(c) = \{(u, v) \in \Xi : u \neq c\}$. Let $X_2$ be a set of constraints of the form $X_2 = \{\bar{x}_2(d) : d \in \mathbb{N}\}$, where $\bar{x}_2(d) = \{(u, v) \in \Xi : v_d = 1\}$. Denote $X = X_1 \cup X_2 \subseteq 2^{\Xi}$.

Let $r \in (0, \frac{1}{2})$. Given $\epsilon \in (0, 1)$ and a probability measure $\mu$ on $X$, consider the following chance-constrained optimization problem:

$$\min_{\substack{u=(u_1,u_2) \\ v=(v_0,v_1,\ldots)}} \quad u_1 + u_2 + \sum_{d=0}^{\infty} r^{d+1} v_d \quad \text{such that} \quad \left\{ \begin{array}{l} v_d = 1 \quad \forall d \in \mathbb{N} \cap [u_1 + u_2, \infty), \\ \mu(\{x \in X : (u, v) \in x\}) \geq 1 - \epsilon. \end{array} \right. \tag{3}$$

---

[9] More formally, if $\boldsymbol{x} = (x_1, \ldots, x_m)$, then $\boldsymbol{x}_s = (x_{i_1}, \ldots, x_{i_n})$ for some $i_1 < \ldots < i_n$ such that $\{x_{i_1}, \ldots, x_{i_n}\} \subseteq X_s$ and for all $j \in [m] \setminus \{i_1, \ldots, i_n\}$, $x_j \notin X_s$.

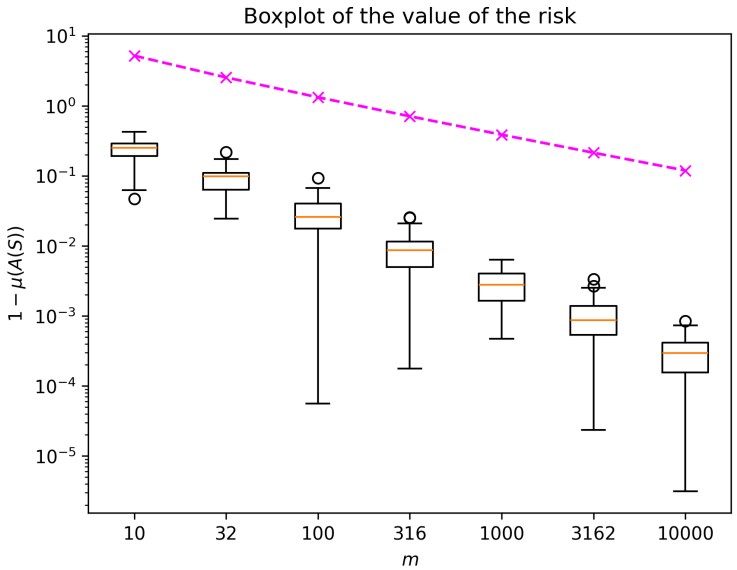

Figure 2: Numerical experiments for the scenario decision algorithm in Example 9, using the probability distribution $\mu$ described in the paragraph below Example 9 with $p = 0.25$. Box plots of the value of the risk $1 - \mu(A(\boldsymbol{x})))$, as $\boldsymbol{x} \sim \mu^m$ i.i.d., for different values of $m$ and doing $M = 50$ experiments for each value of $m$. We see that the risk decreases monotonically (in average and in maximum value) as $m$ increases. The purple line represents the theoretical upper bound (obtained from Theorem 1 and Proposition 6) on the 0.95-th quantile of the value of the risk $1 - \mu(A(\boldsymbol{x}))$. We see that it is indeed above the empirical values.

Solving equation 3 can be approached using scenario optimization. Concretely, given $m$ samples $x_1, \ldots, x_m$ from $X$, consider the scenario optimization problem:

$$\min_{\substack{u=(u_1,u_2) \\ v=(v_0,v_1,\ldots)}} u_1 + u_2 + \sum_{d=0}^{\infty} r^{d+1} v_d \quad \text{such that} \quad \begin{cases} v_d = 1 & \forall d \in \mathbb{N} \cap [u_1 + u_2, \infty), \\ (u, v) \in x_i & \forall i \in [m]. \end{cases} \tag{4}$$

Given $m \in \mathbb{N}$ and $(x_1, \ldots, x_m) \in X^m$, let $(u^\star, v^\star)(x_1, \ldots, x_m)$ be the optimal solution of equation 4.[10] Define the scenario decision algorithm $A : X^* \to 2^X$ by $A(\boldsymbol{x}) = \{x \in X : (u^\star, v^\star)(\boldsymbol{x}) \in x\}$. ◁

We ran numerical experiments on the scenario optimization problem in Example 9 (equation 4). For that, we let $\nu$ be the geometric distribution on $\mathbb{N}$ with success rate $p \in (0, 1)$.[11] Then, we let $\mu$ be the probability measure on $X$ defined by for all $(c_1, c_2) \in \mathbb{N}^2$, $\mu(\{\bar{x}_1(c_1, c_2)\}) = \frac{1}{2}\nu(\{c_1\})\nu(\{c_2\})$ and for all $d \in \mathbb{N}$, $\mu(\{\bar{x}_2(d)\}) = \frac{1}{2}\nu(\{d\})$.

For different samples sizes $m \in \mathbb{N}$ (see horizontal axis in Figure 2), we aimed to estimate the value of $\mu(A(\boldsymbol{x}))$ when $\boldsymbol{x} \in X^m$ is sampled i.i.d. according to $\mu$. For that, we computed $M = 50$ experiments for each value of $m$. The results are reported on Figure 2. We see that the risk decreases monotonically (in average and in maximum value over the $M$ experiments) as $m$ increases. This suggests that the scenario decision algorithm in Example 9 is PAC.

We now prove formally that the scenario decision algorithm in Example 9 is PAC by using the slow internal growth property. For that, we show that its internal growth function is upper bounded by $O(2^{\sqrt{m}})$. Additionally, we show that it is lower bounded by $\Omega(2^{\sqrt{m}})$. Hence, the internal growth function is super-polynomial, but sub-exponential. By Remark 3, the former will imply that the algorithm does not satisfy the sufficient conditions in Section 5.

**Proposition 6.** *The scenario decision algorithm $A$ in Example 9 satisfies $\tau(m) \in 2^{\Theta(\sqrt{m})}$, namely,*

---

[10]In case of ties, take the optimal solution $(u, v)$ with smallest lexicographic order.
[11]That is, for all $k \in \mathbb{N}$, $\nu(\{k\}) = (1-p)^k p$.

*i) for all $m \in \mathbb{N}$, $\tau(m) \leq 2^{2\sqrt{2m}}$;*

*ii) for all $d \in \mathbb{N}$, there is $m \in \mathbb{N} \cap [d, \infty)$ such that $\tau(m) \geq 2^{\sqrt{m-1}}$.*

*Proof.* 1) Proof of i): We show that for any finite set $S \subseteq X$, it holds that $\rho(S) \leq 2^{3\sqrt{m}}$ where $m = |S|$, which will conclude the proof.

For that, fix a finite set $S \subseteq X$, and let $m = |S|$. Let $C = \{c \in \mathbb{N}^2 : \bar{x}_1(c) \in S\}$ and $D = \{d \in \mathbb{N} : \bar{x}_2(d) \in S\}$. Let $k$ be the smallest integer such that $\{(c_1, c_2) \in \mathbb{N}^2 : c_1 + c_2 \leq k - 1\} \subseteq C$. It holds that for all $\boldsymbol{x} \in S^*$, $u_1^\star(\boldsymbol{x}) + u_2^\star(\boldsymbol{x}) \leq k$. Hence, for all $\boldsymbol{x} \in S^*$ and $d \in \mathbb{N} \cap [k, \infty)$, it holds that $v_d^\star(\boldsymbol{x}) = 1$, so that $\bar{x}_2(d) \in A(\boldsymbol{x})$. This implies that $|\{A(\boldsymbol{x}) \cap N_1 : \boldsymbol{x} \in S^*\}| \leq 2^k$. Furthermore, it holds that $|\{u^\star(\boldsymbol{x}) : \boldsymbol{x} \in S^*\}| \leq |C| \leq m$. This implies that $|\{A(\boldsymbol{x}) \cap N_2 : \boldsymbol{x} \in S^*\}| \leq m$. Hence, $\rho(S) \leq m2^{\sqrt{k}}$. Finally, note that $k^2/2 \leq |C| \leq m$. This shows that $\rho(S) \leq m2^{\sqrt{2m}}$. The latter is smaller than $2^{2\sqrt{2m}}$.

2) Proof of ii): Fix $k \in \mathbb{N}_{>0}$ and let $m = k(k+1)/2 + k$. Let $C = \{(c_1, c_2) \in \mathbb{N}^2 : c_1 + c_2 \leq k - 1\}$ and $D = \{0, \ldots, k-1\}$. Let $S_1 = \{\bar{x}_1(c) : c \in C\}$ and $S_2 = \{\bar{x}_2(d) : d \in D\}$. Define $S = S_1 \cup S_2$. It holds that $|S| = m$. We show that for any set $T_2 \subseteq S_2$, it holds that $S_1 \cup T_2 \in \sigma(S)$. This will conclude the proof since it implies that $\tau(k(k+1)/2 + k) \geq 2^k$, hence $\tau(m) \geq 2^{\sqrt{m-1}}$.

To prove the claim, fix $T_2 \subseteq S_2$ and let $T = S_1 \cup T_2$. Let $\boldsymbol{x}$ be a tuple containing all elements of $T$ and only those ones. We show that $A(\boldsymbol{x}) \cap S = T$. Indeed, it holds that $u_1^*(\boldsymbol{x}) + u_2^*(\boldsymbol{x}) = k$. Thus, for each $x \in X_2$, $x \in A(\boldsymbol{x})$ if and only if $x \in T_2$. Hence, $A(\boldsymbol{x}) \cap S = S_1 \cup T_2$, concluding the proof. $\qquad\square$

**Corollary 2.** *The scenario decision algorithm $A$ in Example 9 is PAC.*

*Proof.* Combine i) in Proposition 6 and Theorem 1. $\qquad\square$

A quantitative upper bound on the 0.95-th quantile of the value of the risk $1 - \mu(A(\boldsymbol{x}))$, when $\boldsymbol{x} \sim \mu^m$ i.i.d., obtained from Theorem 1 and i) in Proposition 6 is represented in Figure 2 for different values of $m$.[12] We see that this upper bound is indeed above the empirical values obtained from the numerical experiments.

**Corollary 3.** *The scenario decision algorithm $A$ in Example 9 has infinite VC dimension and does not admit a compression scheme.*

*Proof.* Combine ii) in Proposition 6 and Remark 3. $\qquad\square$

*Remark* 5. The slow internal growth property of the scenario decision algorithm $A$ in Example 9 could have been proved using the $k$-VC dimension criterion (Theorem 2) too. Nevertheless, this would have not provided the quantitative upper and lower bounds on $\tau$ as in Proposition 6.

*Remark* 6. Corollary 3 could have been proved without using the lower bound on the internal growth function provided in ii) in Proposition 6. We find this lower bound informative, as it demonstrates the existence of scenario decision algorithms for which the internal growth function is super-polynomial but sub-exponential. Nevertheless, below is a sketch of alternative proof of Corollary 3.

*Sketch of alternative proof of Corollary 3.* 1) Infinite VC dimension: This comes from the fact that a set of constraints $S = \{\bar{x}_2(d) : d \in \mathbb{N} \cap [0, m-1]\}$ is *shattered* by $\Xi$, i.e., $\sigma'(S) = 2^S$ (with $\sigma'$ as in Definition 10). Indeed, for any $T \subseteq S$, take $u = (0, m)$ and $v = (v_0, v_1, \ldots)$ defined by

$$v_d = \begin{cases} 1 & \text{if } \bar{x}_2(d) \in T \text{ or } d \geq m, \\ 0 & \text{otherwise,} \end{cases} \qquad \forall d \in \mathbb{N}.$$

It is clear that $(u, v) = (u^\star, v^\star)(\boldsymbol{x})$ for some $\boldsymbol{x} \in X^*$; indeed, take $\boldsymbol{x}$ whose elements consist of those of $T \cup \{\bar{x}_1(c) : c = (c_1, c_2) \in \mathbb{N}^2, c_1 + c_2 \leq m - 1\}$. Since $T$ is arbitrary, this shows that $\sigma'(S) = 2^S$, so that $\tau'(m) = 2^m$. Since $m$ is arbitrary, this implies that the VC dimension of $A$ is infinite (Definition 10).

2) No compression scheme: This comes from the fact that the set of constraints $S = \{\bar{x}_1(c) : c = (c_1, c_2) \in \mathbb{N}^2, c_1 + c_2 \leq m - 1\}$ cannot be compressed with a size smaller than $|S| = \frac{1}{2}m(m+1)$. Since $m$ is arbitrary, no compression scheme exists for $A$. $\qquad\square$

---

[12]The upper bound is obtained as the smallest value of $\epsilon$ such that $2e^{-\epsilon m/4}\tau(2m) \leq 0.05$.

### 7.3 Example 1 revisited

Consider the following modification of Example 1:

*Example* 10 (Path planning revisited). Consider a path-planning problem in a discrete "two-dimensional" environment $\mathsf{Env} = \{0, \ldots, M\}^2$, where $M \gg 1$. The goal is to find a path from $\mathsf{S} = (0,0)$ that goes as close as possible to $\mathsf{T} = (M, M)$. See Figure 3 for an illustration. Let $X = \mathsf{Env} \setminus \{\mathsf{S}\}$. The path can only move horizontally and vertically, i.e., each path $\xi$ is described by a sequence of points $(\zeta_0, \ldots, \zeta_r) \in \mathsf{Env}^{r+1}$ such that $\zeta_0 = \mathsf{S}$ and for all $j \in [r]$, $\|\zeta_i - \zeta_{i-1}\|_1 \leq 1$, where the $L^1$-norm is defined by $\|(u_1, u_2)\|_1 = |u_1| + |u_2|$. A random obstacle is present in the environment, where its random position is denoted by $x \in X$ (we assume that the obstacle is not positioned at $\mathsf{S}$). Given $x \in X$ and a path $\xi \doteq (\zeta_0, \ldots, \zeta_r) \in \mathsf{Env}^{r+1}$, we say that $\xi$ collides with the obstacle in position $x$ if there is $j \in [r]$ such that $\zeta_j = x$.

We consider the following algorithm for finding a path with low collision probability from sampled positions of the obstacle. Assume given $m \in \mathbb{N}$ sampled positions of the obstacle, denoted $(x_1, \ldots, x_m) \in X^m$. Define $w \in \mathbb{N}$ as the largest integer such that at least $w^4$ sampled positions are outside $[0, w-1]^2$, i.e., such that $|\{i \in [m] : x_i \notin [0, w-1]^2\}| \geq w^4$. Then, find a path confined in $[0, w-1]^2$ that goes as close as possible to $\mathsf{T}$ and avoids all sampled positions of the obstacle.[13] Denote the algorithm that returns the path given the sampled positions by $A_c$, and denote the associated abstract scenario decision algorithm by $A : X^* \to 2^X$, defined by $A(\boldsymbol{x}) = \{x \in X : A_c(\boldsymbol{x}) \text{ does not collide with the obstacle in position } x\}$. ◁

We show that the scenario decision algorithm in Example 10 is PAC by using the slow internal growth property.

**Proposition 7.** *The scenario decision algorithm $A$ in Example 10 satisfies $\tau(m) \leq 2^{\sqrt{m}}$.*

*Proof.* The proof is similar to the proof of i) in Proposition 6, so we give only the main ideas. If $S \subseteq X$ has size $m$, then the size $w$ in the definition of $A_c$ in Example 10 is at most $\lfloor m^{1/4} \rfloor$. Since the path is confined in $[0, w-1]^2$, it holds that $\sigma(S) \subseteq \{X \setminus T : T \subseteq [0, w-1]^2\}$. Hence, $\rho(S) \leq 2^{w^2} \leq 2^{\sqrt{m}}$. □

**Corollary 4.** *The scenario decision algorithm $A$ in Example 10 is PAC.*

*Proof.* Combine Proposition 7 and Theorem 1. □

*Remark* 7. The path-planning algorithm in Example 10 can be seen as the "quick start" of a longer-horizon path-planning algorithm. Indeed, it can lead quickly and safely to a location in $[0, w-1]^2$ nearer to $\mathsf{T}$, then a more defensive/conservative sample-based path-planning algorithm (e.g., Brudermüller et al., 2024; Cherukuri et al., 2025) can be used for the rest of the path. See Figure 3 for an illustration.

## 8 Conclusion

This work addresses limitations of existing PAC sufficient conditions for scenario decision algorithms, first highlighted by Berger & Jungers (2025). In particular, we propose a new framework, based on the novel notions of internal projected hypothesis classes and internal growth function, to obtain PAC sufficient conditions that are applicable (i.e., non-vacuous) to a broader class of scenario decision problems. The usefulness of our framework is demonstrated on a challenging numerical example.

For future work, we plan to address the question of how to check the slow internal growth property efficiently. In the numerical experiments of this work, the verification of the slow internal growth property was addressed in an ad-hoc manner, but no systematic approach was provided (note that the lack of a systematic verification approach is also present with other PAC sufficient conditions studied in this work[14]). Leveraging the equivalent criterion in Theorem 2 and combination properties such as the one in Proposition 5, we plan to provide mathematical and algorithmic tools for the systematic verification of the slow internal growth property. We also plan to address the question of the necessity of the property. We resolved this question

---

[13]If there is more than one optimal path, take the shortest one, and then the one with smallest lexicographic order.

[14]We note the exception of convex scenario programs, for which the compression size is equal to the dimension of the decision variable (Calafiore, 2010); or the classification problems using simple shapes (e.g., rectangles, spheres, hyperplanes) for which the VC dimension is well known (Shalev-Shwartz & Ben-David, 2014).

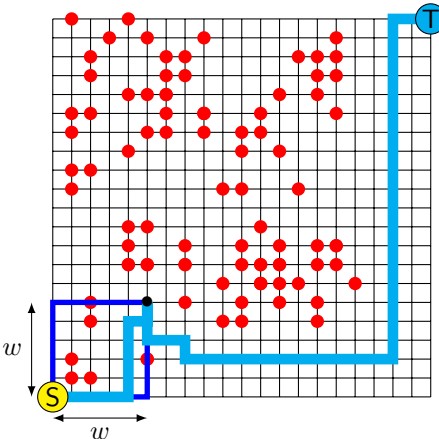

Figure 3: Path-planning problem revisited (Example 10). The sampled positions of the obstacle are in red (note that a same position may be sampled multiple times). The path (light blue) avoids all sampled positions of the obstacle. The value of $w$ and the path from S to $(w-1, w-1)$ (black dot) is computed as explained in Example 10. From $(w-1, w-1)$ to T, any other sample-based path-planning technique—potentially more defensive/conservative (here, a padding of two units of distance from the path to any sampled position in $\mathsf{Env} \setminus [0, w-1]^2$ is required)—can be used.

for the family of stable and monotone scenario decision algorithms, but we plan to address it for the general case in future work.

### Acknowledgments

G. Berger is a FNRS Postdoctoral Researcher. R. Jungers is a FNRS honorary Research Associate. This project has received funding from the European Research Council (ERC) under the European Union's Horizon 2020 research and innovation program under grant agreement No. 864017 (project name: L2C), from the Horizon Europe program under grant agreement No. 101177842 (project name: UniMaaS), and from the ARC (French Community of Belgium) (project name: SIDDARTA). We thank the anonymous reviewers for their insightful comments on the first version of this manuscript.

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
