# OpenReview forum: "The Internal Growth Function: A More General PAC Framework for Scenario Decision Making"
_TMLR — Accepted by TMLR_

### Review · Reviewer_fNKv · 2025-10-23

**Summary Of Contributions:**

The submission is a short note on how to generalize and unify two criteria for PAC learnability of scenario decisions: finite VC dimension and existence of a compression scheme. Each of these two criteria is sufficient for PAC learnability but none is implied by the other. The newly introduced criterion, called slow internal growth, (strictly) generalizes these two criteria (it is implied by each of them but is shown to be more general), is a sufficient condition for PAC learnability, and in addition, is also a necessary condition for a large class of natural scenario decision algorithms (stable and monotone ones).

The short note is written in a concise, neat, and elegant way, while being pedagogical enough (which I can tell because I wasn't familiar with scenario decisions and after investing some time to get familiar with the objects, I could follow perfectly the text and even enjoyed reading it).

**Additional Comments:**

I'm listing my questions and comments in the order of the text.

Page 2, Problem 1 is stated in great generality and it is even unclear that a solution $\xi$ exists. Later, the submission is clear (cf. Problem 2) that Problem 1 may be unfeasible. But at this stage, I was wondering whether I was missing something. Similarly, in Definition 1, I marked to myself that in particular, $A(x_1,\ldots,x_m)$ must belong to $\cap_{i=1}^m x_i$ and I was worried that the latter intersection could be void.

Page 3, Example 1: Is $\overline{x}^{-1}$ a (Borel) measurable map? You would need it for $\mu$ to be well defined. Also, in the definition of $A$ as some argmin: it could well be that no such $\xi$ exists, e.g., if the $m$ obstacles block completely a 'column'. Shouldn't there be some feasability assumptions?

Page 5, Definition 6: I would have expected $m \geq n$ (but $m \leq n$ is considered)

Page 6, Remark 2: Is it $\boldsymbol{x} \subseteq X^\star$ or $\boldsymbol{x} \in X^\star$? Compare to Definition 7, where it is $\boldsymbol{x} \in S^\star$. Also, a pointer to Definition 10 would be useful here.

The proofs of pages 7-8 are elementary and enjoyable to read because 'everything' is in the definitions of Section 3. Some occasional details may be useful, like, adding one or two equations in the middle of page 7 when Chernoff's bound is applied. Also, Footnote 1 could rather be converted in 2 lines of real text, saying that $e^{-\epsilon m /4} \tau(2m)$ can be bounded by something of order $\gamma^m$ with $\gamma = e^{-\epsilon/4} r < 1$ for a well-chosen $r > 1$, thus converges to 0, which entails the existence of $m$.

Page 8, proof of Theorem 1: You do not seem to substitute the bound of Lemma 3 but a smaller bound; indeed, $1-\epsilon/4$ is larger than $e^{-\epsilon/2}$, I believe, but $e^{-\epsilon/4}$ would be fine.

Page 8, Definition 10: Same comment as for Page 6, Remark 2

Section 6: Just a note, the key argument for necessity is thus Berger and Jungers (2025, Theorem 4) but the present short note cleverly relies on it.

Page 13, proof of Corollary 3: When reading it, one may regret that Section 5 did not state in dedicated and formal remarks that for both cases studied therein, the internal growth function was polynomial, and that, combined with the very definition (Definition 9), this only leaves the super-polynomial / sub-exponential case. I find this an interesting insight, that somehow is buried into the proof of Corollary 3. The beginning of Section 5 could also be more specific about this and explain how it will be proved that the new condition is strictly more general.

**Audience:**

Yes

**Audience Explanation:**

Researchers interested in PAC learnability of scenario decisions would be interested in the findings (given the submission offers [almost] a necessary and sufficient condition for that). These researchers belong to several communities but definitely are represented in the machine-learning community, as the bibliography shows (e.g., the 2023 JMLR article by Campi and Garatti).

**Claims And Evidence:**

Yes

**Claims Explanation:**

I carefully read the proofs (except for the ones of Section 7, which I only scanned) and spotted no issue other than occasional clarifications for laymen like me.

**Requested Changes:**

I have no strong such suggestions, the submission is already in excellent shape. I merely have a list of small comments (many due to my prior ignorance of the field, I guess), which I provide below.

---

> ### Author Response · Authors · 2025-12-05
> **Response to review**
>
> Thanks for the careful review and positive feedback.
> Please find below a response to the questions/comments:
>
> >Page 2, Problem 1 is stated in great generality and it is even unclear that a solution $\xi$ exists. Later, the submission is clear (cf. Problem 2) that Problem 1 may be unfeasible. [...]
>
> Thanks for this comment. We do not assume that there exists a solution $\xi$ that satisfies all constraints in $X$. Here, we make a slightly weaker assumption asking that there exists a consistent scenario decision algorithm, which implies that for any finite set of constraints, there is a feasible solution for these constraints (this is more general since an infinite set of sets may have an empty intersection although all finite subsets of sets have a non-empty intersection). The more general setting can be useful in practical situations; for instance if one seeks an upper bound on a random variable with infinite support (there is no finite such upper bound, but there are finite values that are upper bound with probability $1-\epsilon$). We will add a remark to clarify these things.
>
> >Page 3, Example 1: Is $\bar{x}^{-1}$ a (Borel) measurable map? [...] Also, in the definition of $A$ as some argmin: it could well be that no such $\xi$ exists [...]. Shouldn't there be some feasability assumptions?
>
> Thanks for this question. All subsets of $X$ or $X^\*$ considered in the paper are supposed to be measurable with respect to the underlying distribution $\mu$. This includes the sets $\lbrace x\in X : \xi\in x\rbrace$ for all $\xi\in\Xi$, the sets $A(\mathbf{x})$ for all $x\in X^\*$ where $A:X^\*\to2^X$ and the sets $\lbrace\mathbf{x}\in X^* : \mu(A(\mathbf{x})\geq1-\epsilon\rbrace$ for all $\epsilon$. This is a common underlying assumption in scenario optimization (see, e.g., [1], footnote 6). We will add a remark to clarify it.
> As for the feasibility of the “argmin”, this is a good point: we will add the feasibility assumption on $A$; thanks for mentioning it. (Note that since the path is not restricted to the box in Figure 1a, the only way to prevent feasibility is that the boxes completely block the path from $S$ to $T$, e.g., by surrounding $T$ or $S$ or being on $T$ or $S$, which we will exclude by assumption).
>
> [1] Garatti, S., & Campi, M. C. (2025). Non-convex scenario optimization. Mathematical Programming, 209(1), 557-608.
>
> >Page 5, Definition 6: I would have expected $m\geq n$ (but $m\leq n$ is considered)
>
> If all $x_i$ are distinct, then $m\leq n$. But if this is not the case we can have $m>n$ as well. The only requirement is that each sampled constraint $x_i$ appears at least once in $\mathbf{y}$.
>
> >Page 6, Remark 2: Is it $\mathbf{x}\subseteq X^\*$ or $\mathbf{x} \in X^\star$? Compare to Definition 7, where it is $\mathbf{x} \in S^\star$. Also, a pointer to Definition 10 would be useful here.
> >Page 8, Definition 10: Same comment as for Page 6, Remark 2
>
> Thanks for these comment. This is indeed a typo (it should be $\mathbf{x}\in X^*$). We will fix it and add the pointer to Definition 10.
>
> >The proofs of pages 7-8 are elementary and enjoyable to read because 'everything' is in the definitions of Section 3. Some occasional details may be useful, like, adding one or two equations in the middle of page 7 when Chernoff's bound is applied. Also, Footnote 1 could rather be converted in 2 lines of real text, saying that $e^{-\epsilon m/4} \tau(2m)$ can be bounded by something of order $\gamma^m$ with $\gamma = e^{-\epsilon/4}r<1$ for a well-chosen $r > 1$, thus converges to 0, which entails the existence of $m$.
>
> Thanks for the positive feedback and the suggestions for improving the clarity. We will include them in the next versions of the paper.
>
> >Page 8, proof of Theorem 1: You do not seem to substitute the bound of Lemma 3 but a smaller bound; indeed, $1-\epsilon/4$ is larger than $e^{-\epsilon/2}$, I believe, but $e^{-\epsilon/4}$ would be fine.
>
> Thanks for this comment. There is indeed a typo in the first display equation of the proof of Theorem 1: the beforelast term should read $2(1-\epsilon/4)^m\tau(2m)$.
>
> >Section 6: Just a note, the key argument for necessity is thus Berger and Jungers (2025, Theorem 4) but the present short note cleverly relies on it.
>
> Yes, the key argument is Berger and Jungers (2025, Theorem 4). We will add this clarification.
>
> >Page 13, proof of Corollary 3: When reading it, one may regret that Section 5 did not state in dedicated and formal remarks that for both cases studied therein, the internal growth function was polynomial, and that, combined with the very definition (Definition 9), this only leaves the super-polynomial / sub-exponential case. I find this an interesting insight, that somehow is buried into the proof of Corollary 3. The beginning of Section 5 could also be more specific about this and explain how it will be proved that the new condition is strictly more general.
>
> Thank you for these good suggestions: we will include them in next versions of the paper.

---

> > ### Comment · Reviewer_fNKv · 2025-12-05
> > **Thanks for the answers**
> >
> > I thank the authors for their answers and I have no further comment or question.

---

### Review · Reviewer_HGTy · 2025-10-28

**Summary Of Contributions:**

### Summary

In classic classification, we learn a hypothesis $h$ from sampled data so that it generalizes - i.e., it has low misclassification risk; in scenario decision making (SDM), a “scenario” is a realized random constraint (modeled here as the set of decisions that satisfy it), and the goal is to compute from sampled scenarios $S$ a decision $x=A(S)$ whose population violation risk is small with high confidence. This paper studies PAC guarantees for such scenario decision algorithms. They note that while extending VC/Rademacher and compression can yield guarantees, these routes are sufficient but not necessary PAC in SDM.

To go beyond them, the paper introduces a strictly (supported by examples in Sec 7) more general complexity notion - the Internal Growth Function (IGF) in Sec. 3, extending the classic growth function to the SDM setting. They further prove slow (sub-exponential) IGF implies PAC in Sec 4; prove existing VC/compression conditions imply slow IGF in Sec 5; and prove IGF corresponds exactly to k-VC dimension (finite k-VC $\Leftrightarrow$ slow IGF) in Sec 5. Moreover, it is shown in Sec 6 that for stable monotone algorithms, slow IGF is also necessary, while the necessity direction for general SDM remains open.

### Strengths

The paper addresses key limitations of existing PAC guarantees for SDAs, clarifying what is sufficient versus necessary and closing some gaps in our understanding of PAC conditions for scenario decision making. I appreciate that several examples are provided to help ground the abstract concepts and guide the reader through the messages.

### Weaknesses (mostly presentation/clarity)

Despite the examples, the presentation remains dense and contains multiple abstraction layers. The work would benefit from more bridging sentences to smooth the transitions.

**Audience:**

Yes

**Audience Explanation:**

PAC guarantee is an important way to understand the fundamental learnability of a problem. While PAC theory has been extensively studied for classic supervised learning, it is far less developed for SDM, where key gaps remain. This paper is interesting because it directly targets those gaps. It introduces an idea (the Internal Growth Function) that extends the classic “growth function” to SDM, links it to known tools (like VC and compression), and shows when it gives guarantees even where existing methods don’t. Hence, it grounds PAC guarantees for SDM in a framework that is pretty well-aligned with classical learning theory.

**Claims And Evidence:**

Yes

**Claims Explanation:**

Mostly yes, but several arguments would benefit from brief supporting explanations, e.g., in Examples 3, 4, and 5, even if they are relatively straightforward.

**Requested Changes:**

1) **Add brief explanations to Examples 3–5.**
   Even if the steps are simple, including 1–2 sentences of brief explanations can be helpful for the readers.

2) **Brief introduction to $X\subseteq 2^\Xi$ (Sec. 2.1).**
   This can look a bit abrupt at first glance. It could be helpful to motivate by something like “A realized constraint can be represented as the set of decisions that satisfy it. ...” Then give the formal definition.

3) **Motivate the abstraction (Sec. 2.2).**
   It could be helpful to start with why it is useful. E.g., abstraction removes the explicit decision parameterization and directly tracks which constraints the output satisfies.

4) **Clarify the intuition in Example 9.**
   This example somehow does not look immediately natural to me. Please add further intuition for why this problem cannot be proved PAC using the criteria in Section 5, but can be proved PAC using the slow internal growth criterion.


### Minor
1. Typo: in the proof of Lemma 2, there's a bracket of indicator function, you should change from } to ].

---

> ### Author Response · Authors · 2025-12-05
> **Response to review**
>
> >Mostly yes, but several arguments would benefit from brief supporting explanations, e.g., in Examples 3, 4, and 5, even if they are relatively straightforward.
>
> Thanks for the careful review, positive feedback and suggestions for improvement. We will add explanatory and bridging sentences to improve the clarity, especially when introducing and using abstract concepts.
>
> >Add brief explanations to Examples 3–5.
> Even if the steps are simple, including 1–2 sentences of brief explanations can be helpful for the readers.
>
> Thanks for this comment. We will add explanations to help the reader to understand these examples and the concepts they illustrate.
>
> >Brief introduction to $X\subseteq 2^\Xi$ (Sec. 2.1).
> This can look a bit abrupt at first glance. It could be helpful to motivate by something like “A realized constraint can be represented as the set of decisions that satisfy it. ...” Then give the formal definition.
>
> Thanks for this comment. We will add such explanation/motivation sentences to make the presentation smoother.
>
> >Motivate the abstraction (Sec. 2.2).
> It could be helpful to start with why it is useful. E.g., abstraction removes the explicit decision parameterization and directly tracks which constraints the output satisfies.
>
> Thanks for this suggestion. We will include it in the paper.
>
> >Clarify the intuition in Example 9.
> This example somehow does not look immediately natural to me. Please add further intuition for why this problem cannot be proved PAC using the criteria in Section 5, but can be proved PAC using the slow internal growth criterion.
>
> Thanks for this comment. We will add explanations of why this example does not satisfy the criteria in Propositions 2 (finite VC) or 3 (compression) without using the internal growth function being super-polynomial (as we do currently in the paper). This should indeed help the reader to get more intuition about the example since our proof via the internal growth function (although having the advantage of being short and leveraging Proposition 6) lacks intuition.
>
> *Glimpse of the proof*: For the VC dimension, it comes from the fact that a set of constraints $S=\lbrace\bar{x}_2(d) : d=1,\ldots,m\rbrace$ can be “shattered”, i.e., $\sigma’(S)=2^S$ (with $\sigma’$ as in Definition 10). Since $m$ is arbitrary, this implies that the VC dimension of $A$ is infinite (see Definition 10).
> For the compression: Note that a set of constraints $S=\lbrace\bar{x}_1(c) : c\in\mathbb{N}^2, c_1+c_2\leq m\rbrace$ cannot be compressed with any size $<m^2$. Since $m$ is arbitrary, the compression size is infinite.

---

### Review · Reviewer_ofkR · 2025-11-30

**Summary Of Contributions:**

The paper proposes a new PAC framework for scenario decision-making based on the notion of internal growth, an adaptation of classical growth functions to the scenario setting. It shows that sub-exponential internal growth is sufficient to guarantee PAC learnability and provides explicit sample-complexity bounds. The framework generalizes existing PAC results based on VC dimension and compression schemes, and introduces the k-VC dimension as an equivalent characterization of slow internal growth. The theory is demonstrated on examples, including a two-stage scenario optimization problem.

**Audience:**

Yes

**Audience Explanation:**

The paper addresses a fundamental question in scenario optimization and learning theory: when sample-based decision algorithms admit PAC guarantees. By introducing a more general framework that extends beyond classical VC-dimension and compression-based analyses, the work is directly relevant to researchers in statistical learning theory. The new internal growth notion, as well as the demonstrated applicability to non-standard scenario optimization problems, are appreciated.

**Claims And Evidence:**

Yes

**Claims Explanation:**

The main claims are supported by rigorous theoretical analysis. The authors provide formal definitions of internal growth, prove that slow internal growth implies PAC guarantees with explicit sample-complexity bounds, and establish connections to existing VC-dimension and compression-based criteria. The illustrative examples, including the two-stage scenario optimization problem, demonstrate cases where proposed method applies, providing concrete validation of analysis usefulness.

**Requested Changes:**

* While slow internal growth is the core sufficient condition of the paper, the manuscript does not discuss a practical way to verify it for a given scenario decision algorithm. Practical guidelines for checking this property for new algorithms are missing, which limits the applicability of the framework. Providing clearer check would be valuable.

* The necessity result is restricted to stable monotone algorithms. More discussion of whether this condition might extend beyond this subclass would be appreciated.

* While the synthetic two-stage example effectively demonstrates the advantages of the proposed approach, adding more realistic or varied application examples (e.g., from control or robotics) would further support the practical significance of the framework.

---

> ### Author Response · Authors · 2025-12-05
> **Response to review**
>
> Thanks for the careful review and positive feedback.
> Please find below a response to the questions/comments:
>
> >While slow internal growth is the core sufficient condition of the paper, the manuscript does not discuss a practical way to verify it for a given scenario decision algorithm. Practical guidelines for checking this property for new algorithms are missing, which limits the applicability of the framework. Providing clearer check would be valuable.
>
> Thanks for this comment. At this point, we do not have a systematic way of checking the slow internal growth property, even for specific subclasses of problems. *This is a question that we plan to address in future work.* We note that for compression or finite VC dimension, except for some specific classes of problems (such as convex scenario optimization or SVM), there are no general systematic ways of checking these properties either.
> Also, note that the $k$-VC dimension provides an alternative way of checking the slow internal growth property (Theorem 2), possibly enabling the use of approaches from VC theory (because of the link between the $k$-VC dimension and the VC dimension).
>
> Furthermore, it is a common practice in machine learning to design algorithms so that they satisfy a given sufficient generalization condition (e.g., in [2] the authors modify existing learning algorithms such as Neural Network training to enforce compression). The slow internal growth property being more general, it offers more flexibility in the design of these algorithms, thereby allowing for improved performance. To give an example of meta-algorithm satisfying the slow internal growth property, suppose that there is a sequence of subsets $X_1\supseteq X_2\supseteq X_3\supseteq \ldots$ with $X_1=X$. Also, for each $k=1,2,\ldots$, let $A_k:X_k^\*\to\Xi$ be a scenario decision algorithm with VC dimension $k$.$^1$ A scenario decision “meta-algorithm” $A:X^\*\to\Xi$ can be defined as follows. Given a sample set $\mathbf{x}=(x_1,\ldots,x_m)$, let $k(\mathbf{x})$ be the largest integer $k$ such that all and at least $k^2$ elements of $\mathbf{x}$ are in $X_k$. The output of $A(\mathbf{x})$ is defined as the output of $A_{k(\mathbf{x})}(\mathbf{x})$. This meta-algorithm $A$ can be shown to satisfy the slow internal growth property. This offers a lot of flexibility in combining existing scenario decision algorithms to improve performance.
>
> [2] Paccagnan, D., Campi, M., & Garatti, S. (2023). The Pick-to-Learn algorithm: Empowering compression for tight generalization bounds and improved post-training performance. Advances in Neural Information Processing Systems, 36, 18165-18185.
>
> $^1$ To give an example (inspired from Example 9), each $X_k$ could allow for one more degree of freedom (i.e., one more decision variable). In this way, our approach allows us to tackle infinite-dimensional scenario optimization problems. See also the response to the last comment for an illustration on a path-planning problem.
>
> >The necessity result is restricted to stable monotone algorithms. More discussion of whether this condition might extend beyond this subclass would be appreciated.
>
> Thanks for this comment. For the moment, despite some effort spent on this question, it is still unclear to us whether the necessity result extends to other classes of algorithms. This is a question that we plan to keep investigating in future work. We will clarify this in the paper.
>
> >While the synthetic two-stage example effectively demonstrates the advantages of the proposed approach, adding more realistic or varied application examples (e.g., from control or robotics) would further support the practical significance of the framework.
>
> Thanks for this very good comment. We will add an example inspired from control/robotics. The example will consist in a path-planning problem and will be a variation of Example 1. More precisely, the setting will be similar to Example 1, but the goal will be to minimize the distance between $T$ and $\zeta_H$ (see Figure 1a) where $H$ (the number of viapoints) is a decision variable as well. The length of the path is then $\leq\ell H$ where $\ell$ is the maximum distance between two consecutive viapoints. If the path is piecewise linear (i.e., linear between the viapoints), the VC dimension of the problem with $H$ viapoints is $\leq cH$ for some constant $c$ (proof omitted). Then, we decide the value of $H$ depending on the sample set $\mathbf{x}$ to satisfy the slow internal growth property. For instance, if $h(\mathbf{x})$ is the smallest integer $h$ such that a sampled obstacle in $\mathbf{x}$ is *reachable* with a path with $h$ viapoints, then we let $H$ be given by the largest integer smaller than $h(\mathbf{x})+\sqrt{\lvert\mathbf{x}\rvert}$.

---

### Decision · Action_Editor_TngP · 2026-01-13

**Recommendation:** Accept as is

**Audience:**

Yes

**Audience Explanation:**

It is clear that this will be an interesting work for the learning theory community.

**Claims And Evidence:**

Yes

**Claims Explanation:**

This is a learning theory work studying PAC bounds of certain function classes. Authors showed that for the proposed scenario decision-making setup, prior PAC bounds can be degrade or even vacuous. A new notion, termed internal growth function, is studied and is shown to provide meaningful characterization of learnability.